# Challenges in Local Allergic Rhinitis Diagnosis, Management, and Research: Current Concepts and Future Perspectives

**DOI:** 10.3390/medicina59050929

**Published:** 2023-05-11

**Authors:** Mohamad Mahdi Mortada, Marcin Kurowski

**Affiliations:** Department of Immunology and Allergy, Medical University of Łódź, 92-213 Łódź, Poland; mohamad.mortada@umed.lodz.pl

**Keywords:** local allergic rhinitis, nasal mucosa, nasal secretions, immunoglobulin E (IgE), nasal allergen provocation test, allergen immunotherapy

## Abstract

Local allergic rhinitis (LAR) is diagnosed based on the presence of clinical symptoms such as rhinorrhea, sneezing, and nasal itching using negative skin prick testing and serum IgE assessment. Several novel studies have shown that it is possible to use the assessment of nasal sIgE (specific immunoglobulin E) secretion as an additional diagnostic criterion for local allergic rhinitis. Additionally, allergen immunotherapy is a promising—albeit still not fully assessed and evaluated—future method of managing patients with LAR. In this review, the historical background, epidemiology, and main pathophysiological mechanisms of LAR shall be presented. Additionally, we address the current state of knowledge based on selected articles regarding the assessment of the local mucosal IgE presence in response to exposure to such allergens as mites, pollen, molds, and others. The impact of LAR on quality of life as well as the possible options of management (including allergen immunotherapy (AIT), which showed promising results) will then be presented.

## 1. Introduction

Local allergic rhinitis is a subtype of allergic rhinitis (AR) that affects a great percentage of individuals who were historically diagnosed as having non-allergic rhinitis (NAR). So far, those individuals have undergone the classic management approach for NAR patients. Although helpful in alleviating their symptoms, such management schemes deprive them of specific allergy treatment. Noninfectious rhinitis can be classified as allergic or nonallergic based on clinical history and a skin prick test (SPT) and/or serum IgE findings. Allergists and other specialists are often confronted with diagnostic challenges posed by patients with a clear history of typical nasal symptoms elicited by allergen exposure, but in whom the IgE-mediated etiology of disease cannot be confirmed using any standard diagnostic tool. Despite apparently clear and convincing history, both skin prick testing and serum-specific IgE results may be negative regarding the suspected culprit allergen. Such a situation raises doubts and substantially limits the options regarding the further management of the patient.

However, it has become clear that these systemic assessments do not always reflect allergic inflammation occurring in the nasal mucosa and limited to that compartment. Hence, the term “local allergic rhinitis”, acronymized as LAR, was first proposed in 2009 by Rondón et al. [1]. AR and LAR share clinical features typical of any kind of rhinitis, i.e., rhinorrhea, sneezing, and nasal itching. However, in LAR, systemic IgE sensitization cannot be ascertained either through SPT or through assessment of allergen-specific IgE in serum. In the presence of clearly defined symptoms correlated with defined allergen exposure, finding evidence of local mucosal IgE synthesis is important for elucidating the mechanism responsible for clinical features in this distinct population of rhinitis subjects. Proving the presence of local IgE production in the nasal mucosa (in the absence of systemic IgE sensitization) enables the identification of subjects with local allergic rhinitis.

Therefore, LAR can be strongly suspected in the presence of the following clinical features: typical clinical symptoms (rhinorrhea, sneezing, and nasal itching), negative SPT, absence of allergen-specific IgE in the serum, and a positive nasal allergen provocation test (NAPT).

Additionally, the term “dual allergic rhinitis” (DAR) has been coined by Eguiluz et al. [2]. DAR can be applied to describe the coexistence of the following features: perennial rhinitis symptoms, nasal allergen-specific reactivity to both perennial and seasonal allergens, and positive SPT results only for seasonal allergens. In other words, in DAR subjects two types of nasal reactivity to different allergens can coincide, the one confirmed through IgE presence and the other without ascertaining the presence of allergen-specific IgE on the skin or in the serum. Terms used in current discussions regarding subtypes of rhinitis and used throughout this review are listed in Table 1.

## 2. Aim

This review paper aims to acquaint the readers with current insights into the pathogenesis and clinical manifestation of LAR. Special focus will be put on the local production of IgE in the nasal mucosa and the challenges associated with the investigation of this phenomenon. In addition, details regarding inflammatory processes (particularly type 2 inflammation) lying behind the development of the clinical features of LAR will be addressed. The current state of knowledge, research gaps, and future research perspectives will be presented. Finally, perspectives on the clinical management of patients with confirmed LAR will be discussed.

## 3. Literature Search

This review included a literature search using Medline, Embase, and PubMed. Search terms included the following: local allergic rhinitis, nasal provocation test, local specific IgE, entopy, allergen immunotherapy, nasal mucosa, and IgE production. A total of 53 articles were identified initially, excluding duplicates. Subsequently, original articles and review articles were chosen independently by both authors based on their titles and abstracts. Case reports and all types of articles written in languages other than English were excluded. The choices made by either author were then compared and confronted. Finally, a consensus was reached, and 38 articles were selected (32 original articles and 6 reviews). No restriction regarding the publication date was applied.

## 4. Beginnings of the Research on LAR

Although both the concept and the definition of LAR were proposed relatively recently, studies focusing on the local presence of allergen-specific IgE in nasal mucosa have been conducted over the past several decades. In 1970, Tse et al. [3] detected ragweed pollen-specific IgE in nasal secretions of ragweed-sensitized rhinitic subjects. Five years later, Huggins et al. [4] confirmed for the first time the local mucosal production of allergen-specific IgE, reporting the presence of IgE antibodies specific to house dust mites in the nasal secretions of patients with typical AR symptoms but with negative skin results. These findings strongly supported the assumption that the local production of IgE has been taking place in the nasal mucosa in subjects with rhinitis symptoms, but in whom the systemic presence of IgE cannot be ascertained.

A few years later, Platts-Mills [5] showed that in most grass pollen allergic subjects allergen-specific IgG, IgA, and IgE were present in the serum and in the nasal mucosa, with a higher level of pollen-specific antibodies in the nasal mucosa. Later, Powe et al. detected the presence of specific IgE antibodies in nasal secretions in three non-atopic idiopathic rhinitis subjects and in eight allergic subjects and proposed the term “entopy” to define the concept of localized allergic reaction [6].

In 1994, Sensi et al. [7] showed that the increase in house dust mite (HDM)-specific IgE levels following re-exposure is much faster at the local mucosal level than in the serum. In 2016, an interesting study by Gelardi et al. [8] was published in which the presence of allergen-specific IgE (asIgE) in the nasal mucosa was assessed in patients with allergic rhinitis, patients with non-allergic rhinitis, and in healthy controls. In their study, the local presence of IgE was confirmed in all three groups of patients. Based on these findings, the authors suggested that local IgE production may be a part of a spontaneous, non-specific immune response to environmental factors. They advised caution in establishing LAR diagnosis, even if asIgE is present in the nasal mucosa.

Figure 1 presents the main historical checkpoints of the research regarding LAR, starting in 1947 with Samter et al. and progressing through the coining of the term “local allergic rhinitis” in 2009 [9].

## 5. Epidemiology and Diagnostic Challenges of LAR

For a long time, LAR was considered a rare and uncommon disease. Nowadays, the actual prevalence of LAR remains incompletely assessed. The reported frequency of LAR varies between studies, and this is due to differences in studied populations and suspected causative agents (allergens), as well as the research methodology employed. Importantly, diagnostic algorithms and pathways should make provisions for the possibility of IgE-mediated sensitization limited to the nasal mucosa and include such a possibility in the diagnostic approach. However, the possible limitations of such an approach should be considered, including, firstly, the requirements of the experienced healthcare personnel for the acquisition of nasal secretions, and, secondly, the influence of technical procedures, such as the dilution of the samples, which can impact the outcomes of sIgE measurement. The proposed diagnostic algorithm to be implemented in the management of allergic rhinitis subjects is presented in Figure 2.

LAR has been ascertained in between 21% and 62.5% of adult subjects in whom rhinitis symptoms have been noted [11,12]. In the pediatric population, reported LAR frequency varies from 3% to 66.7% [9,13]. As is reviewed in detail below, various allergens may be of significance in the pathogenesis of LAR symptoms, with notable differences between populations and regions. 

The prevalence of newly defined and described dual allergic rhinitis (DAR) is estimated at 11.6% and 82.1% among the pediatric and adult populations, respectively [2,14,15]. The possibility of the coexistence of AR and LAR in one subject, as it is observed in DAR, further hinders the process of designing and carrying out epidemiological studies on LAR.

Overall, the prevalence of LAR varies between populations and geographical regions. Table 2 summarizes data from LAR studies in which epidemiological aspects have been considered.

## 6. Pathomechanisms of LAR

The pathophysiology of local allergic rhinitis has not been fully elucidated yet [26]. Several studies have been undertaken in order to explain and provide more insights into the pathophysiological mechanisms of LAR.

In 2001, in research conducted by Powe et al., the assessment of the nasal mucosa of idiopathic rhinitis subjects through in situ hybridization showed activation of type 2 inflammatory mechanisms and an increase in eosinophils and mast cell count when compared with normal subjects [27].

Eguiluz-Gracia et al. shared in their review article an interesting overview of previous investigations carried out regarding the different pathophysiological mechanisms possibly underlying LAR. Among others, basophil activation tests (BATs) performed on subjects with suspected LAR show a positive result in a considerable proportion of cases [28].

As mentioned above, a type 2 inflammatory response has been demonstrated in the nasal mucosa of LAR patients, and the immediate activation of mast cells and eosinophils in LAR patients was observed. The release of inflammatory mediators such as tryptase and eosinophilic cationic protein (ECP) in LAR patients was positively assessed [1].

Several proteins have been considered and assessed as potential LAR biomarkers; however, no conclusions have been reached in this regard, and the quest for markers that could help define the LAR phenotype continues. Proteins studied in the context of LAR characterization are listed in Table 3.

One of the main challenges in research on LAR pathogenesis is to ascertain the presence of specific locally produced IgE in the nasal mucosa. Over the past decade, several research groups have successfully proved the presence of local IgE production at the level of the nasal mucosa in subjects suspected of having LAR [31], based on the criteria we mentioned earlier. In the study by Powe et al., an increase in IgE+ B cells was demonstrated in the nasal mucosa of idiopathic rhinitis subjects [27]. Additionally, wortmannin (a PI3k blocker that inhibits IgE-dependent activation) inhibited basophil activation during BAT [28].

It is also worth mentioning that in some cases, LAR can overlap with NARES (nonallergic rhinitis with eosinophilia syndrome), an entity that was first described more than 40 years ago. Assessment of nasal eosinophils (nEo) can be used as a preliminary marker which requires a further NAPT to discriminate NARES from LAR [32]. Another marker worth considering is CLC (Charcot–Leyden crystals), the levels of which are significantly higher in NARES subjects than in healthy controls [33]. The differentiation between NARES and LAR is another emerging challenge for clinicians taking care of rhinitis subjects.

## 7. Local Allergic Rhinitis with Sensitization to Specific Allergens—Current State of Knowledge

### 7.1. House Dust Mites

In a study conducted by Rondón et al., 110 subjects (50 patients with perennial NAR (PNAR), 30 with perennial AR (PAR), and 30 healthy controls) were assessed for several factors, including nasal IgE specific to *D. pteronyssinus*. It was shown that 22% of the subjects within the PNAR group had nasal IgE specific to *D. pteronyssinus* [34].

In another study by Gelardi et al., a total of 41 subjects (15 with AR, 12 with NAR, and 14 healthy controls) were recruited. Nasal IgE assessments for several allergens, including dust mites, were performed on all subjects in addition to clinical assessment, SPT, serum IgE assay, nasal endoscopy, and nasal cytology. The results showed the presence of nasal IgE specific to mites in 60.7% of the nonallergic groups (NAR and control) [8].

In a similar study conducted by Powe et al. in 2003, 33 subjects were recruited (11 with atopic rhinitis, 10 with non-atopic rhinitis, and 12 healthy controls) and assessed for the presence of grass pollen- and dust mite-specific IgE in the nasal mucosa [6]. The results were positive for specific IgE in the nasal mucosa of 8 out of the 11 atopic subjects. Additionally, grass pollen-specific IgEs were detected in one third of the samples from the non-atopic individuals, while all the samples from the non-atopic individuals were negative with regard to mite-specific IgE.

In a large study by Bożek et al., 219 patients were recruited. The assessment included SPT, serum total specific IgE, and a NAPT with common aeroallergens, as well as the investigation of the nasal presence of specific IgEs before and after the challenge. The results showed that 46 patients could be classified as having LAR, 88 had AR, and 85 could be diagnosed with NAR. *Dermatophagoides pteronyssinus* was found to be the main sensitizing allergen in the LAR group (29 patients). There was also a significant increase in nasal IgE production induced by the nasal allergen challenge [11].

### 7.2. Pollen Allergens

A study conducted in Poland by Krajewska-Wojtys et al. included 121 subjects with NAR (ages between 12 and 18 years old). The patients were assessed with SPT, serum, and nasal-specific IgE and a NAPT for several pollen allergens (grass, mugwort, and birch). A group of control subjects were assessed as the remaining subjects. The results showed the presence of LAR in 16 patients allergic to grass, 6 patients allergic to mugwort, and 9 patients allergic to birch pollen. The results of the NAPT and serum sIgE assessments were concurrent with each other within the analyzed groups [18].

In another study conducted in Spain, 61 patients with symptoms of seasonal rhinitis and negative SPT and serum-specific IgE were recruited. Their assessment included a specified questionnaire and measurement of their serum-specific IgE and a NAPT with timothy grass pollen. The response to the NAPT was monitored using an assessment of nasal symptoms, acoustic rhinometry, and a determination of the sIgE, tryptase, and eosinophil cationic protein in the nasal secretions. The employment of the above-listed assessment tools revealed the presence of seasonal LAR in 37 patients without allergic sensitization features ascertained through the methods used during standard management [17].

Other studies addressing the local nasal allergic reaction to pollen allergens include the previously cited works of Gelardi et al. [8] and Powe et al. [6]. The former proved the local presence of IgE specific to the pollen allergens of parietaria (67.9%), olive (26.8%), cypress (15.4%), and grasses (38.1%) in the nasal mucosa of the non-allergic subjects (NAR and controls). In the latter study, grass pollen-specific sIgE was detected in three subjects from the non-atopic IR group and eight subjects from the atopic IR group.

### 7.3. Molds and Animal Allergens

In a study aiming to assess the local allergic response to mold allergens, a total of 60 subjects (40 subjects with NAR and 20 healthy controls) were recruited. The subjects were then divided into two groups. In the first group, a NAPT with *Aspergillus fumigatus* allergen was performed, whereas in the second group, a mixture of *Alternaria alternata/Cladosporium herbarum* allergens was used in the challenge. Mold-specific IgE and tryptase in the nasal secretions of both groups did not differ from those of the healthy controls. However, among the NAR patients, eight challenged with *A. fumigatus* and nine challenged with Alternaria/Cladosporium had positive NAPT results [19].

In another study, 84 patients with perennial rhinitis symptoms, a negative SPT, and no serum IgE specific to common allergens were recruited. NAPTs with several allergens, including Alternaria and cat allergen, were performed, followed by the measurement of specific IgE in the nasal secretions. The results were positive for *Alternaria* in two patients, and for cat allergen in one patient [24].

As has been revealed in the above-mentioned studies, the presence of allergen-specific IgEs can be ascertained not only in subjects with rhinitis symptoms of confirmed allergic etiology, but also in those who are classified initially as having non-allergic rhinitis. The presence of nasal IgE production in LAR subjects was more pronounced for mite and pollen allergens than for molds or animal dander. However, this might be due to the fact that more research has been carried out on local mite and pollen allergies than on allergens originating from other sources. More research must be undertaken to consolidate the previous findings as this could reveal new possibilities in the implementation of specific allergen immunotherapies in management protocols as a method of treatment for LAR patients.

## 8. LAR Clinical Course and Prognosis

LAR is a relatively new clinical entity, and therefore little is known about its long-term course or prognosis. The main hitherto unresolved issues regarding the clinical course of LAR include (1) changes in LAR symptom intensity over a long period of time; (2) the probability of its conversion to systemic allergic rhinitis; and (3) the frequency of development of asthma in subjects presenting LAR symptoms. Rondón et al. [35] attempted to fill this knowledge gap by conducting a 10-year follow-up study on a cohort of 176 LAR patients and 115 matched healthy controls.

The evolution of the clinical presentation of LAR included a worsening of symptoms with a tremendous increase in the need to be admitted to the emergency department with the development of asthma, and a decrease in quality of life. The worsening of symptoms became more significant after the fifth year and progresses throughout the 10 years. At the end of the observation period, 17 patients (9.7%) initially classified as having LAR developed systemic atopy. Among the healthy controls, the development of systemic atopy features was observed in nine subjects (7.8%), though this was not significantly different from the LAR group. LAR has also been identified as a considerable risk factor for the development of asthma, the diagnosis of which was confirmed in slightly more than 30% of the subjects after the 10-year observation period. Conjunctivitis was another comorbidity that was observed during the observation follow-up. However, the increase in its prevalence after 10 years was not statistically significant (52.3% at baseline vs. 61.9% in the tenth year).

## 9. Possible Options for LAR Management

Local allergic rhinitis can be classified as a well-differentiated clinical disease with a present but low rate of development of systemic manifestation and a natural evolution towards worsening and an increased risk of developing asthma. One also must emphasize here that with the low outcomes of the traditional methods of treatment that are normally used in the management of this disease, such as educating patients, allergen avoidance, and the usual medications used as symptom relievers, there is a need to implement new management options which can provide patients with symptom relief and improve the quality of their daily life. Allergen immunotherapy (AIT), which has an established position in the management of allergic rhinitis with systemic IgE-sensitization features, has not been evaluated in LAR patients to the extent that its inclusion as a recommended management option is permitted.

AIT has been tackled in several studies, and these have shown promising results in that the decrease in the intensity of daily LAR symptoms results in a reduction in the negative impacts on quality of life. AIT is a highly effective and safe approach to management which also provides long-term benefits for patients, even after the discontinuation of the treatment [36].

Bożek et al. [37] assessed the efficacy of sublingual immunotherapy (SLIT) for HDM and concomitant asthma in 32 patients with LAR. The patients were divided into 2 groups: 17 patients received SLIT with a *D. pteronyssinus* and *D. farinae* mixture, while 15 patients received a placebo. The following parameters were included in the assessment: total rhinitis score (TRSS), total asthma symptom score (TASS), combined total symptom score (TSS), total medication score (TMS), and forced expiratory volume in the first second (FEV1). Only the patients who completed the entire trial were included in the final analysis, which showed that the patients who received the SLIT experienced a considerable decrease in all the mentioned score values and an increase in the mean FEV1 compared with the placebo group.

In another trial conducted by the same group [38], 28 LAR patients were recruited. Of these, 15 patients received subcutaneous allergen immunotherapy for birch, while the remaining 13 patients were given a placebo. The following parameters were then assessed: symptom medication score, serum-specific IgE, serum-specific IgG4, and nasal IgE specific to Bet v 1 (birch pollen allergen). The AIT did not cause any systemic reactions in the active group and was well tolerated. After 24 months of treatment, the results showed a decrease in the symptom medication score (SMS) in the active group when compared with the placebo group. There was also an increase in the IgG4 and a decrease in the nasal-specific IgE in the active treatment group compared with the placebo group.

## 10. Conclusions

Local allergic rhinitis, although a relatively newly described clinical entity, is gaining increasing attention and interest in the allergist community. The ambiguity of its clinical presentation, the lack of established guidelines, and the low accessibility of diagnostic tools contribute to delays in diagnosis and the suboptimal management of LAR patients.

The most considerable knowledge gaps concerning local allergic rhinitis can be summarized as follows:Further assessment of its clinical course over a long period of time is necessary, with stress upon the conversion to systemic atopic disease and the risk of the development of other respiratory and allergic comorbidities.More knowledge is needed regarding local allergic rhinitis caused by allergens other than house dust mites and pollen.The phenotypic characterization of LAR, including an assessment of the presence and role of inflammatory mediators at the local mucosal level, should be undertaken.Reliable assessment methods for detecting allergen-specific IgE in nasal secretions should be developed.The efficacy of specific allergen immunotherapies in the management of LAR patients should be assessed.

## Figures and Tables

**Figure 1 medicina-59-00929-f001:**
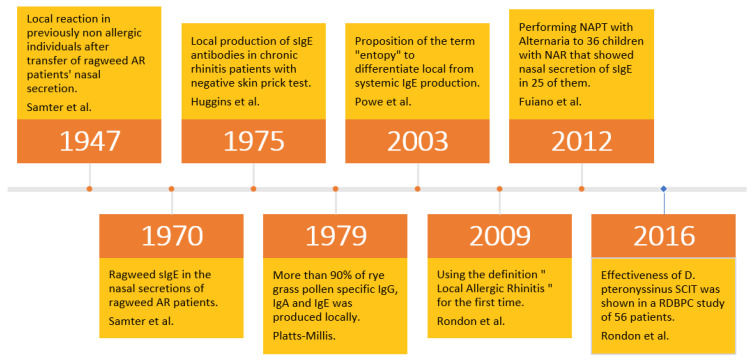
Main checkpoints on the LAR research pathway (adapted from Beken et al. [9]).

**Figure 2 medicina-59-00929-f002:**
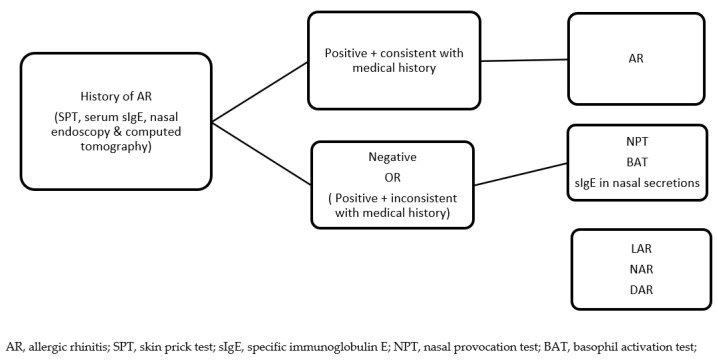
Diagnostic algorithm of chronic rhinitis (adapted from Krzych-Fałta E et al. [10]).

**Table 1 medicina-59-00929-t001:** Nomenclature applied in currently discussed rhinitis classifications.

AR	Allergic rhinitis
LAR	Local allergic rhinitis
DAR	Dual allergic rhinitis (coexistence of AR and LAR)
NAR	Non-allergic rhinitis
NARES	Non-allergic rhinitis with eosinophilia
PNAR	Perennial non-allergic rhinitis

**Table 2 medicina-59-00929-t002:** Selected studies assessing the prevalence of LAR.

Study	Number of Subjects	LAR Prevalence	Allergens	Age	Ethnicity
Zicari et al. [16]	20	66.7%	HDM Grass pollen	Range 6–12	Italy
Rondon et al. [12]	32	65%	Grass pollen	Mean age 41 ± 18	Spain
Blanca Lopez N et al. [17]	61	61%	Gras pollen		Spain
Krajewska-Wojtys A et al. [18]	121	16.6% 8.9% 5.9%	Grass pollen Birch Artemisia	Range 12–18	Poland
Demirtürk M. et al. [19]	40	66.6% 69.2%	*Aspergillus* *fumigatus* *Alternaria alternata and Cladosporium herbarum*	Mean age 36.4 ± 10 36.4 ± 11.5	Turkey
Rondon et al. [20]	428	25.7%	D.p (*Dermatophagoides pteronyssinus)* *Alternaria alternata Olea europea* Grass pollen Dog and cat epithelia	Range 14–68	Spain
Bozek et al. [21]	621	17.6%	D.p Grass Pollen	Mean age 17.6 ± 4.8	Poland
Tantilipikorn P et al. [22]	62	24.2%	D.p	Mean age 36.1 ± 10.4	Thailand
Jang and Kim [23]	110	10.9%	D.p	Range 11–69	South Korea
Krajewska-Wojtys A et al. [24]	84	25%	D.p Alternaria Cat allergen	Mean age 29.4 ± 9.1	Poland
Cheng et al. [25]	147	8.2%	D.f (*Dermatophagoides farinae*)	Range 9–78	China
Buntarikckpornpan et al. [13]	54	3.7%	D.p	Range 8–18	Thailand

**Table 3 medicina-59-00929-t003:** Proteins studied in the context of participation in LAR pathogenesis.

Study	Proteins
Rondon et al. [1]	Tryptase, ECP
Zicari et al. [16]	IL-5, TSLP
Kim et al. [29]	IL-13, IL-5, IL-10, TGF-β
Baumann R et al. [30]	IL17, MCP-4, Eotaxin-3

ECP: eosinophil cationic protein; IL-5/13/10/17: interleukin 5/13/10/17; TSLP: thymic stromal lymphopoietin; TGF-β: transforming growth factor beta.

## Data Availability

Not applicable.

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
