# Peer review of "Challenges in Local Allergic Rhinitis Diagnosis, Management, and Research: Current Concepts and Future Perspectives"

_medicina, 2023, doi:10.3390/medicina59050929_

Round 1
Reviewer 1 Report
The article is of great value. I have no concerns or suggestions to make. Language is appropriate and methods are according to the rules of evidence-based medicine. Authors made a great job.
Author Response
Thank you for the positive feedback
Reviewer 2 Report
The review provides a clear overview of the main topics covered in local allergic rhinitis (LAR), including the diagnosis, pathophysiology, and management of LAR. It also highlights the importance of recent research on the assessment of nasal sIgE secretion and the potential of allergen immunotherapy as a treatment option.
MINOR COMMENTS:
An graphical to illustrate of the pathophysiology of the disease or mechanism of action of the disease would likely enhance the overall impact of the article. I would recommend the author to consider relevant and informative figure in the article. The article has few grammatical and punctuation errors, I also recommend the author to perform a careful grammar check before submitting the article.
The article has few grammatical and punctuation errors, I recommend the author to perform a careful grammar check before submitting the article.
Author Response
We have corrected grammar and punctuation errors. Regarding graphical presentation of LAR pathogenesis, we did not foresee it initially, since the main scope of our article was to highlight state of knowledge with regard to different allergens, which - in our opinion - discerns our review form others in the filed.
Reviewer 3 Report
The review paper "Challenges in local allergic rhinitis diagnosis, management and research: current concepts and future perspectives" by Mortada and Kurowski is an interesting and thorough article describing what LAR is and how it has been studied throughout the decades. It summarizes well what is known about the prevalence and underlying mechanisms and takes a look into possible treatments in the literature.
The review article is well-structured and logical and it points out areas that have been studied more and areas that are less defined. It was informative to see the whole history of research on LAR laid out in the article, going even back to 40's. Most cited articles are rather recent.
However, I have some comments that I believe would improve the paper:
It is clear from the text that the frequency of LAR among people suffering from rhinitis symptoms varies greatly. The authors mention that employed research methods are one explanation to this. Furthermore, they conclude that one of the knowledge gaps regarding LAR is the development of reliable assessment methods of detecting allergen specific IgE in the nasal secretion. I would like to see some reviewing of these methodological issues in more detail: what kind of differences are there actually in the research methods that might lead to different end results in the frequency of LAR? is there a known technical issue or issues in the detection of sIgE in nasal secretion that might compromise result reliability or make studies more difficult to perform? I interpret from the text that detecting sIgE might be challenging to perform, and became curious about what factors might contribute to this.
In Table 1, the authors have selected some LAR studies where epidemiological aspects have been considered. I would ask some clarification on what is meant by these epidemiological aspects: the review discusses many articles in sections 6.1, 6.2 and 6.3 that do not seem to be in Table 1. Could the authors explain in more detail why the articles in table 1 have been selected for presentation there? I found it a bit confusing that the text above the table discusses the large variation in LAR frequency and cites the extremes, but then they are not found in the table beneath, except for reference 12. Could the table summarize even all articled discussed in the review that study the frequency of LAR? If not, then please explain what is meant by the epidemiological aspects as a criteria for presentation.
I would ask the authors to carefully check the use of abbreviations and remove multiple explanations for them in the text, as well as making sure that all abbreviations are explained on every occasion on first mention: for example, sIgE in the Abstract, SPT in the Introduction, DP and DF in Table 1, FEV1 in Possible options for LAR management.
In Figure 1, sIgE is mentioned in two boxes - does the one in the first one (leftmost box) actually mean sIgE in serum, while the one on the middle one in the right refers to nasal secretion sIgE?
The review discusses several different forms of rhinitis: there is AR, LAR, DAR, NAR, PNAR, NARES... It gets confusing easily. Suggestion: could there be an info box, where all of these are collected with a brief explanation for each?
Language is mostly good but needs some revising (grammar in some places, the use of punctuation marks, some typos).
Author Response
Dear Reviewer,
Please find below explanations of amendments included into aour manuscript as per your suggestions,
- Regarding the first point, we added a part in the "Epidemiology and diagnostic challenge of LAR", with a small clarification on possible limitations for the method regarding sIgE assessment in nasal secretions.
- Regarding the table , we've added the most important remaining studies mentioned in our review, so the table now includes the prevalence ranges from lowest to highest.
- We have included the nomeclature summary in a table directly below the introduction
- Regarding the remaining parts, abbreviations have been added as recommended and grammatical and punctuation errors have been corrected.
- Clarification has been done regarding information regarding sIgE on figure 2.